# Exposure to Roundup and Antibiotics Alters Gut Microbial Communities, Growth, and Behavior in *Rana berlandieri* Tadpoles

**DOI:** 10.3390/biology12091171

**Published:** 2023-08-25

**Authors:** Melissa Villatoro-Castañeda, Zachery R. Forsburg, Whitney Ortiz, Sarah R. Fritts, Caitlin R. Gabor, Camila Carlos-Shanley

**Affiliations:** 1Department of Biology, Texas State University, 601 University Dr., San Marcos, TX 78666, USA; melvcc@gmail.com (M.V.-C.); zforsburg@archbold-station.org (Z.R.F.); whitney.ortiz11@gmail.com (W.O.); fritts.sarah@txstate.edu (S.R.F.); carlos-shanley@txstate.edu (C.C.-S.); 2Archbold Biological Station, 123 Main Dr., Venus, FL 33960, USA; 3Department of Molecular Microbiology and Immunology, The University of Texas at San Antonio, One UTSA Circle, San Antonio, TX 78249, USA

**Keywords:** amphibians, bacteria, gut microbiome, pollutants, tadpoles, glyphosate

## Abstract

**Simple Summary:**

Gut microbiomes can influence host health and fitness. Pollutants, including antibiotics, tend to alter microbiomes. We examined the role of an undisturbed gut microbiome on tadpole health and morphology in the Rio Grande Leopard frog, *Rana berlandieri*. We exposed tadpoles to four treatments (1) control: clean water, (2) Roundup^®^: the active ingredient is glyphosate, the main herbicide used in the United States, (3) antibiotic cocktail, to disrupt the natural microbiome of the tadpoles, and (4) combination: Roundup and antibiotic cocktail. We found that the gut microbial community significantly changed across treatments. Tadpoles in the antibiotic and combination treatments were least active and the smallest compared to the other treatments. Our results provide evidence that the gut microbial communities of tadpoles are sensitive to herbicides and antibiotics, which may have an impact in host phenotype and fitness via altered behavior and growth. This study provides important insights for conservation of amphibians and into the consequences of current agricultural practices.

**Abstract:**

The gut microbiome is important for digestion, host fitness, and defense against pathogens, which provides a tool for host health assessment. Amphibians and their microbiomes are highly susceptible to pollutants including antibiotics. We explored the role of an unmanipulated gut microbiome on tadpole fitness and phenotype by comparing tadpoles of *Rana berlandieri* in a control group (1) with tadpoles exposed to: (2) Roundup^®^ (glyphosate active ingredient), (3) antibiotic cocktail (enrofloxacin, sulfamethazine, trimethoprim, streptomycin, and penicillin), and (4) a combination of Roundup and antibiotics. Tadpoles in the antibiotic and combination treatments had the smallest dorsal body area and were the least active compared to control and Roundup-exposed tadpoles, which were less active than control tadpoles. The gut microbial community significantly changed across treatments at the alpha, beta, and core bacterial levels. However, we did not find significant differences between the antibiotic- and combination-exposed tadpoles, suggesting that antibiotic alone was enough to suppress growth, change behavior, and alter the gut microbiome composition. Here, we demonstrate that the gut microbial communities of tadpoles are sensitive to environmental pollutants, namely Roundup and antibiotics, which may have consequences for host phenotype and fitness via altered behavior and growth.

## 1. Introduction

Animals support microbial communities, providing a unique ecosystem for microorganisms with complex microbiome-host interactions [1]. The microbiome community can be shaped by host and environmental factors, such as environmental surfaces, exposure to pathogens, and horizontal/vertical bacterial transmission [2,3,4] and may impact host fitness and health [5]. The diversity of these microbial communities influences the host’s metabolism, immunity, and nutrient uptake [1,6,7,8]. Disruption of the gut microbiome community via environmental pollutants can lead to metabolic disorders and greater risk of bacterial infections [9,10,11].

The host-associated gut microbiota helps regulate development and growth [12,13,14] and is especially important during growth in juvenile stages [15,16]. An undisturbed gut microbiome can promote growth through improved digestion and metabolism [17,18]. In addition, microbes can manipulate host behavior [19] to their own advantage [20] and microbes influence the production or synthesis of signaling molecules that control both food consumption and food choice [1]. Changes in the microbiome influence host food selection and grooming; reproduction may also control the abundance and composition of the microbiome [21,22]. Together, these factors affect overall fitness and growth.

Exploring the core microbiome community (i.e., the prevalent taxa dominating the microbiome) among host organisms of the same species provides insights into the naturally occurring communities and eco-evolutionary dynamics [11,23,24]. Microbial communities associated with amphibian skin and gut interact with the environment and provide defense against parasites and pollutants [25]. The capacity of a host to harbor certain bacterial communities may vary with the ecological environment, because there is a strong relationship between the host condition and environmental interaction with the microbial community residing within the host [1]. Indeed, host microbiomes are different from environmental microbiomes [26,27], and the microbiome can transition if the host’s environment is disturbed. Therefore, studies focusing on the responses to environmental shifts of both the host and their microbiomes are needed to understand the active role microbial communities play in host health and fitness [28]. 

A vast array of contaminants reach aquatic environments, where amphibians inhabit, breed, and develop, modifying the microbiota and altering amphibian susceptibility to parasites [29,30,31]. Two common contaminants include pharmaceuticals (e.g., antibiotics) from humans and livestock and pesticides from farming and other human uses [11,32]. These environmental stressors result in dysbiosis of host microbiomes, leading to loss of microbial community diversity, higher abundance of pathogenic species, and loss of beneficial bacteria [33,34,35]. In amphibians, the functional change of the gut microbiota may be more sensitive to environmental disturbance than the amphibian hosts themselves [36], and because their gut and skin microbiota are habitat-specific, environmental factors might be mediating bacterial community structure [37]. Glyphosate, the active ingredient in Roundup^®^ herbicides, is the main herbicide component used in the United States, and its use has increased worldwide [32,38]. This herbicide inhibits the synthase of 5-enolpyruvylshikimate-3-phospate, an enzyme found in plants and bacteria. Glyphosates and other herbicides alter the microbiota of many animals [39,40]. Glyphosate significantly changes the skin microbiota in cricket frog larvae, *Acris blanchardi* [41], however not much is known about how glyphosate and antibiotics affect the gut microbiome of amphibians [42]. Understanding of the initial core microbiome composition and functionality of the naturally occurring microbiomes is useful to understand amphibian declines [28,43] and to potentially relating findings to other species, including humans exposed to these pollutants. 

We explored how disturbance via Roundup, and in combination with antibiotics, alters gut bacterial community composition. Cuzziol et al. [42] found that *Rhinella arenarum* tadpoles exposed to glyphosate-based herbicides had greater taxa diversity whereas tadpoles exposed to Ciproflaxin antibiotics had decreased intestinal bacterial diversity. We hypothesized that a low, yet environmentally relevant, concentration of Roundup (in Roundup ^®^ Ready-to-use weed and grass killer III) and/or exposure to antibiotics would alter the community composition of the gut microbiome of Rio Grande Leopard frog, *Rana berlandieri*, tadpoles. The gut microbiome of *R. berlandieri* larvae has not been studied. Studies of gut microbiomes in Leopard frogs, *Rana pipiens*, found that the dominant genus is Proteobacteria and Firmicutes in tadpoles [44] and Bacteroides in adults [45,46]. They also found anaerobic bacteria such as butyrogenic and acetogenic, protozoans and several ciliates [46]. Negative effects of Roundup exposure in tadpoles of the *Rana* genus have been reported by Relyea [47,48,49], including larval toxicity, lethality, and growth of deeper-tails suggesting an activation of developmental pathways used for antipredator responses. Similarly, Gabor et al. [50] found deeper tails in Roundup-exposed *Rana berlandieri* tadpoles. We hypothesized that tadpoles with disturbed microbiome (via exposure to Roundup, antibiotic, and combination treatments) would show differences in activity, growth, food consumption, and mortality than those that were not exposed. Exploring the gut microbiome of anuran populations provides insights into the fitness trade-offs between survival and growth, owing to detrimental alterations to the environment caused by anthropogenic factors. 

## 2. Materials and Methods

We collected three egg clutches of *Rana berlandieri* on 24 February 2020 from a stream located in Spring Lake Reserve, San Marcos, Texas (first egg clutch: 29°54′02.9″ N, 97°56′36.1″ W; second and third egg clutches: 132 cm apart, 29°54′01.4″ N, 97°56′39.7″ W); leaving half of each clutch undisturbed at the site. *Rana berlandieri*, the Rio Grande Leopard frog, inhabits streams, rivers, side pools, pools along arroyos, springs, and stock tanks in several environments such as grasslands, semiarid areas, mountainous regions, and woodlands [51]. 

We brought egg clutches into the laboratory at Texas State University and housed them in 3 L plastic tanks with 2.7 L of stream water collected from the site until the eggs hatched. When tadpoles were free-swimming (approximately one week after collection), we haphazardly mixed tadpoles from each egg clutch and reared them in 32–6 L plastic tanks filled with 2.7 L of spring water, following Gabor et al. [50]. They were left undisturbed under a natural light cycle (via a large window) at room temperature (24 °C). We fed tadpoles *ad libitum* with agar food blocks (a mixture of spirulina powder and fish flakes (ISO flake food TetraMin) in an agar medium); four food block squares were added every four days and any uneaten food was removed. We monitored tadpoles daily and changed water once a week and as needed. We reared the tadpoles until Gosner stage 25 [52] and then randomly assigned them to new 3 L treatment tanks (*n* = 6 tadpoles per tank, 8 tanks per treatment). We allowed tadpoles to acclimate to the new tanks for two days before exposing them to the treatments.

We exposed *R. berlandieri* tadpoles (*n* = 192) to four treatments for three weeks: (1) Roundup treatment (environmentally relevant concentration of 1.47 mg a.e/L of glyphosate in Roundup ^®^), (2) antibiotic treatment (to alter the natural microbiome from the tadpoles): spring water with antibiotic cocktail (30 mg/L enrofloxacin, 13.3 mg/L sulfamethazine, 2.67 mg/L trimethoprim, 5000 µg/L streptomycin, and 5000 I.U./L of penicillin), and (3) combination treatment: same concentrations as above of Roundup and antibiotic cocktail, and a (4) control group of tadpoles in spring water. For the Roundup and combination treatments, we mixed 45 L spring water with 1.14 mL of Roundup^®^, doubling the concentration reported by Gabor et al. [50]. We housed the tanks with tadpoles inside a sterilized incubator. The antibiotic cocktail was used by Knutie et al. [53] to disrupt the early-life microbiota of Cuban tree frog tadpoles (*Osteopilus septentrionalis*) and did not present lethality. We wore new gloves for each tank throughout the experiment to avoid cross contamination. Temperature (24 °C) and light cycle (12 h light/12 h darkness) remained constant through the exposure period inside the incubator. On day 10, we performed a complete water change for all treatments and re-dosed with the same quantities described above. 

We photographed of each tadpole at the beginning, end, and every fourth day of exposure. To do this, we gently caught all tadpoles per tank at once in a clean net and took a picture at the water surface level, with a ruler for scale. We measured the dorsal body area, as this provides a good estimate of body condition [54]. Therefore, we use dorsal body area as an assessment of fitness. Dorsal body area was determined using ImageJ [55]. We counted the number of consumed agar food blocks, removed sediment and uneaten food, and added four new agar food blocks per tank every four days. On day 11 of exposure, we placed the tanks on a white surface with web cameras mounted above each tank, to measure tadpole activity. We acclimated tadpoles for 20 min after we moved them to the white surface, and then recorded them for 60 min. We quantified activity (total distance moved cm) from the videos using Noldus ^®^ EthoVision. 

We collected the guts of two tadpoles per treatment tank (*n* = 16/treatment) after three weeks of exposure to the treatments. We euthanized tadpoles by placing them on an ice bath for an hour. We removed the abdominal plate of each larvae using sterile bistoury blades and forceps. We placed each sample in individual sterile Eppendorf tubes and stored them at −20 °C. DNA extraction, PCR, purification, quantification, and sequencing of gut samples follow the methods described below. 

To explore longer term (four months) recovery from the treatments, we used four tadpoles from each tank (*n* = 32/treatment) that we did not dissect after the exposure period ended. First, we thoroughly rinsed the tanks with deionized water to ensure no treatments remained in the tanks and added new spring water. Then tadpoles were kept under the same laboratory conditions previously described. We also continued to measure their dorsal body area during the recovery period, record the amount of food consumed, and mortality per tank. We changed tank water every 15 days. Once tadpoles reached Gosner stage 40, we euthanized them with a 1% benzocaine overdose. We stored the specimens in 75% ethanol. After four months of recovery, we euthanized all tadpoles that had not reached Gosner stage 40.

### 2.1. Microbiome Assay

We performed total DNA extraction for 16S rRNA gene amplicon. We analyzed samples following Gontang et al. [56] and Gabor et al. [57] for gut microbiome analysis. In summary, we extracted DNA from the samples by using Pure Link™ Microbiome DNA Purification Kit. We then performed two rounds of PCR to obtain DNA fragments with specific primers for barcode analysis. We confirmed amplification by using gel electrophoresis with a Thermo Scientific GeneRuler 1kb Plus DNA Ladder. We conducted a PCR purification with ExoSap–IT PCR Product Cleanup kit, and a final DNA concentration measure with Qubit dsDNA BR assay kit. For sequencing, we diluted samples to obtain 40ng/µL and combined them in one sterile Falcon tube. Amplicon libraries were sequenced with the paired-end Illumina MiSeq platform at the Texas State University.

### 2.2. Statistical Analysis: Dorsal Body Area, Food Consumption, and Death Rates through Exposure and Recovery Periods

We calculated the mean dorsal body area from the pictures. Because variances were not equal across treatments, we performed a repeated measures nested ANOVA with a rank transformation on dorsal body area as the response variable, treatment, and day of exposure as fixed factors, and tank number as a random factor nested in day of exposure. We used the residual maximum likelihood (REML) method and calculated Tukey-adjusted comparisons (alpha = 0.05) to detect which treatments were significantly different. We then plotted dorsal body area across treatments through the different time points. We performed the same analysis with rank transformation using the exposure measurements, recovery measurements, consumed food during exposure, and death rates during recovery. We analyzed the dorsal body area from the starting point of the experiment versus the final measurements of the exposure period by performing the same analysis but with a logarithmic transformation of dorsal body area (cm^2^) because it was not normally distributed. We performed the same analysis for consumed food during the recovery period. 

### 2.3. Behavior Analysis

Using Noldus EthoVision^®^ software, we analyzed tadpole activity using distance moved (cm) which was not normally distributed, so we performed a logarithmic transformation. We conducted a mixed effects linear model with treatment as a fixed factor, and tadpole nested into tank number as a random factor. We used an ANOVA to test for differences across treatments. 

### 2.4. Microbiome Analysis

We analyzed the FASTA sequencing results by following the DADA 2 [58] pipeline in R for sample filtering and identification. We used the SILVA 138.1 database [59] to assign taxonomy to the filtered bacterial amplicon sequence variants (ASVs). All samples were processed together. We removed samples under 1000 sequencing reads. We rarefied the relative abundance data (seed: 999, permutations: 999) and conducted a permutated redundancy analysis (RDA) to analyze the differences in composition and relative abundance between treatments. Using the rarefied data, we calculated alpha diversity indexes using Hill numbers (with the vegetarian package in R, [60] and created an Alpha diversity profile. We also calculated community turnover between treatments using rarefied Hill numbers. Alpha diversity is best described by Hill numbers [61,62,63,64,65].

We used the Microbiome package in R [66] to describe the core microbiome of the treatments (prevalence 50%) and the average most relative abundant phylum and genera that characterizes each microbiome. We performed a multivariate analysis with linear models to analyze differences in microbial genus found per treatment using the Galaxy platform 1.01. We further analyzed these data by individual Kruskal-Wallis tests to determine which treatments differ. 

## 3. Results

### 3.1. Dorsal Body Area and Food Consumption

There were no significant differences in the dorsal body area of tadpoles exposed to our treatments at the beginning of the exposure (ANOVA: F_1,28_ = 1.81, *p* = 0.17, Figure 1A day 0). However, at the end of the three weeks of exposure, there were significant differences in the dorsal body area of the tadpoles across treatments (Rm nested ANOVA: F_1,148_ = 7.33, *p* < 0.0001, Figure 1A). Control (*p* < 0.001) and tadpoles exposed to Roundup^®^ (*p* < 0.001) grew the largest compared to antibiotic- and combination-exposed tadpoles. Control- and Roundup-exposed tadpoles did not differ significantly (*p* = 0.07) neither did antibiotic- and combination-exposed tadpoles (*p* = 0.64). None of the tadpoles died during the exposure period.

There were significant differences in the number of consumed agar food blocks across treatments (Rm nested ANOVA: F_3,124_ = 4.02, *p* = 0.009, Figure 1B). More specifically, tadpoles in control treatment consumed more food than tadpoles in Roundup (*p* = 0.001), antibiotic (*p* < 0.0001), and combination (*p* < 0.0001) treatments. Antibiotic- and combination-exposed tadpoles did not consume significantly different amounts of food blocks compared to each other (*p* = 0.56). 

There were significant differences in dorsal body area at the end of the recovery period across treatments (Rm nested ANOVA: F_3,853_: 9.10, *p* < 0.001, Figure 1C). Tadpoles that were previously in the control and Roundup treatments presented the greatest dorsal body area, but these two treatments were not significantly different from each other (*p* = 0.73). Tadpoles in the combination and antibiotic treatment had the smallest dorsal body area but were not significantly different from each other (*p* = 0.11). Because of an incubator malfunction, all tanks were reared outside the incubator for the last month (20 July–19 August 2020), at ~24 °C with a natural light cycle. Nine control tadpoles and nine Roundup-exposed tadpoles reached Gosner stage 40, whereas four tadpoles from the combination treatment and one from the antibiotic treatment reached this stage (out of 48 tadpoles per each treatment). We found no significant differences in survival across treatments during this period (Rm nested ANOVA: F_3,853_ = 0.05, *p* = 0.99).

### 3.2. Behavior

There were significant differences across treatments in tadpole activity during exposure (ANOVA: F_3,139_ = 83.62, *p* < 0.0001, Figure 2). Control tadpoles were the most active (Linear mixed model fit by REML: *t*-value = 13.02, *p* < 0.0001); followed by Roundup-exposed tadpoles (*t*-value = 9.94, *p* < 0.0001). There were no significant differences between the activity of combination- and antibiotic-exposed tadpoles (*t*-value = 1.10, *p* = 0.27). However, antibiotic-exposed tadpoles were the least active compared to control and Roundup (*t*-value = 90.26, *p* < 0.0001). 

### 3.3. Gut Microbiome

From Rio Grande Leopard frog tadpole gut microbiome samples (*n* = 60), 16S rRNA amplicon sequencing resulted in 1,091,946 sequences and 813 unique ASVs. After filtering and trimming the sequences, two gut samples (from the combination-exposed tadpoles) were removed from the data. There were significant differences in the gut microbiome composition and relative abundance across all combination of treatments (RDA: F_29_ = 7.523, *p* = 0.001, R^2^ = 0.25, Table 1). 

We plotted the alpha diversity profile across treatments (Figure 3). We did not find significant differences in the species richness across treatments (ANOVA: q = 0: F_29_._50_ = 0.876, *p* = 0.459). We found significant differences in the number of common species (q = 1) and number of very abundant species (q = 2) across treatments (ANOVA: q = 1: F_29,50_ = 6.184, *p* = 0.01; q = 2: F_29,50_ = 8.343, *p* = 0.0001), more specifically between control and antibiotic tadpoles (q = 1: *p* = 0.01; q = 2: *p* = 0.0001), and antibiotic- and Roundup-exposed tadpoles (q = 1: *p* = 0.007; q = 2: *p* = 0.001). We found community turnovers across treatments (Appendix A).

We described the detection threshold by the core gut microbiome of tadpoles from each treatment (Figure 4A–D). The most prevalent ASV per group were classified to the genus level when possible. The number of genera constituting each core gut microbiome varied: control (*n* = 18), Roundup (*n* = 16), combination (*n* = 16), and antibiotic (*n* = 2). 

## 4. Discussion

Anthropogenic pollutants alter the microbial communities on the skin and in the gut of amphibians [67,68,69,70]. However, more research is needed to understand how altered microbial communities affect fitness, phenotype, and behavior. Both Roundup and an antibiotic cocktail altered the composition and relative abundance of the gut microbiome, as well as the growth and behavior of Rio Grande Leopard frog tadpoles. Similar to other studies of vertebrates including amphibians, the most abundant phyla of intestinal bacteria were Proteobacteria and Firmicutes [71,72,73]. The higher abundance of more pathogenic bacteria, Bacteroidota and Firmicutes, in the combination treatment is associated with the breakdown of organic matter and could result in these tadpoles being most susceptible to lower health via altered immune systems owing to the lower quality bacterial communities. Tadpoles exposed to Roundup had a higher relative abundance of pathogenic bacteria (such as *Legionella* sp, see below) in their gut microbiome compared to the control tadpoles and were less active than control tadpoles. Antibiotic- and combination-exposed tadpoles were the smallest and slowest compared to control tadpoles and did not present pathogenic bacteria in high relative abundance but had low relative abundance of *Xanthobacter. Prevotella_9* was found in high abundance only in the combination-exposed tadpoles. These results support the hypothesis that gut microbial communities of anuran larvae are sensitive to environmental pollutants such as Roundup and more so when their gut bacteria were already disrupted, and this in turn, can affect amphibian health. 

Glyphosate alone negatively affects anuran fitness, specifically by altering growth and development [68,74]. Despite finding significant differences in dorsal body area between control and antibiotic groups, we did not find significant differences in dorsal body area between control and Roundup-exposed *R. berlandieri* tadpoles. Roundup induced mortality and growth changes in other tadpole species [75,76,77], though we did not observe this in our study. Even though we did not find significant growth differences between Roundup-exposed and control tadpoles, we observed a trend of tadpoles being exposed to Roundup being smaller, suggesting that Roundup exposure might delay metamorphosis. Slower growth and smaller body size at metamorphosis has long term fitness effects [78,79] and might also affect overwintering survival and later life stages fitness [79,80]. Given that there is a trade-off between survival and growth in tadpoles, our results suggest that tadpoles prioritize allocating more energy in survival than in growth. Despite not observing any significant size differences, we did find that tadpoles exposed to Roundup were significantly less active than control tadpoles. Tadpole activity can be used as a biomarker for toxicity evaluation [81], since a decrease in activity increases predation risk, reduces feeding, and correlates with slower growth [82]. With the observed difference in activity and the trend of smaller growth, our findings suggest that at higher levels than we used, but still environmentally relevant, Roundup can affect the fitness of anurans tadpoles.

Antibiotics and pharmaceuticals are known to reach aquatic ecosystems, potentially causing long-term risks to aquatic and terrestrial organisms associated with such ecosystems [83,84,85]. In fact, one of the antibiotics we used, enrofloxacin, is one of the most used antibiotics in human and veterinary medicine [86], and although it has not been found to cause significant mortality in *Rhinella arenarum* tadpoles [86], it can cause growth and development suppression (in doses higher than 10µg/L) in *R. arenarum* tadpoles [86]. One mechanism by which antibiotics reduce growth and development is by inducing neurotoxicity in tadpoles, which is associated with reduced food intake and reduced activity [86]. Antibiotic-exposed tadpoles consumed the least amount of food during exposure and recovery periods in our study. Another proposed mechanism is that the natural gut microbiome could be directly altered by the antibiotic, inducing significant short-term changes to the microbial community and host health [86]. Therefore, the role of the microbiome in hosts’ energetic gain and the direct relation to body size might change if antibiotic is present in the environment [87,88]. A disturbed gut microbiome could lack symbiotic microbes that produce essential amino acids, vitamins, and short-chain fatty acids which are a pivotal part of host metabolism and nutrient processing, allowing for endocrine pathways that regulate growth [13]. 

We predicted that tadpoles with disturbed microbiomes would be smaller, because an undisturbed microbiome produces greater concentrations of bacterial metabolites that allow an increase in somatic growth rates and fat storage [8]. Here, there were differences in activity, growth, and microbial composition in antibiotic-exposed tadpoles compared to control tadpoles, suggesting that the differences in growth due to antibiotic exposure occurs by a combination of both pathways, through behavioral and microbial community changes. Overall, we did not find significant differences in activity, growth, and gut microbial composition between combination- and antibiotic-exposed tadpoles, suggesting that the effects of the antibiotic cocktail alone were enough to suppress growth, change behavior, and decrease food consumption in *R. berlandieri* tadpoles. 

While our results suggest that Roundup and antibiotic exposure altered amphibian gut microbial communities, further studies are needed to understand if the effects in phenotype and fitness are due to this alteration or to the host’s response to the treatments. The differences in the alpha diversity profile between control- and antibiotic-exposed tadpoles are more noticeable in a high ASV richness, and a high number of abundant ASVs in the antibiotic treatment. Several studies only use diversity in microbial research as a parameter of a stable and healthy microbial community [10], however, Coyte et al. [88] found that a greater richness tends to have a destabilizing effect, suggesting that the changes in the community composition are more relevant than just the diversity of bacteria. This is evident in antibiotic and combination groups because there is a greater bacterial richness comprising the core microbiome of these exposed tadpoles and a high community turn over compared to control tadpoles. The disruption of the natural occurring bacterial communities by Roundup and antibiotics might have facilitated certain bacteria to colonize the gut microbiome and to shift the community structure [87] compared to the control group. We present a clear example of this in the Roundup-exposed tadpoles, where a disrupted microbiome allowed for higher relative abundance of *Legionella* in this treatment only. *Legionella* in tadpoles has been identified in farmland frogs and associated with infectious disease-related pathogens [71]. This genus is associated with the gut microbiome of frogs with diseases, for example diarrheic captive *Rana dybowskii* [89]. Another example of this is in the high relative abundance of *Bosea* and *Xanthobacter* in the Roundup-exposed tadpoles compared to the other treatments. These two genera are in the gut microbiome of *Rana chensinensis* tadpoles exposed to octylphenol, an endocrine-disruptive chemical [90]. These changes provide evidence that the disruption of the natural occurring gut microbiome in *R. berlandieri* tadpoles by Roundup allows for colonization of pathogenic bacteria that might influence the overall health and development of anuran larvae. Despite not finding a high relative abundance of pathogenic bacteria in the antibiotic and combination treatments, these treatments presented the highest ASV evenness. Since we found differences in body condition and behavior, we suggest that a high richness in the bacterial community may be associated with a negative effect in fitness and behavior in anuran tadpoles.

The microbiome contributes to metabolism, energy uptake from food, and defense against pathogens [7,91,92], amongst many other well-documented functions. Because there were differences in the microbial composition and relative abundance across treatments, we can suggest that the growth and behavior differences found in Roundup, antibiotic, and combination treatments compared to control tadpoles, are associated with either a disturbed microbial community and/or to Roundup alone. We monitored changes in the gut microbiome through taxonomic diversity, however functional diversity analyses to monitor the changes in microbial functions after exposure are needed to further understand the effects Roundup and a disturbed microbiome have on the host behavior, growth, and survival.

## 5. Conclusions

A disturbed microbiome has impacts on host health and fitness of *Rana berlandieri* tadpoles. From a management perspective, our results indicate that even low levels of Roundup should not be used during amphibian breeding and larval periods to protect the natural gut microbiome, normal growth rate, and typical behavior in *R. berlandieri* tadpoles. In this study, we did not observe an interaction between antibiotic-induced changes and Roundup exposure. This could have been driven by the relatively high concentration used in our antibiotic cocktail. Future studies should use another antibiotic cocktail or target specific members of the core microbiome of the study species to test the effects of a disturbed microbiome in host health and fitness. 

## Figures and Tables

**Figure 1 biology-12-01171-f001:**
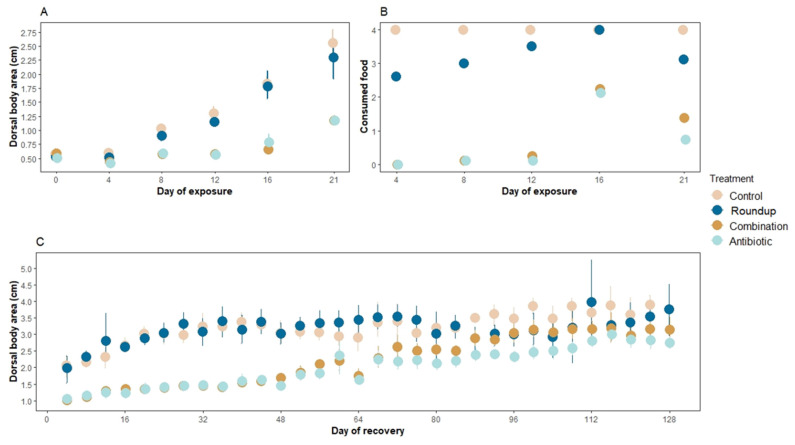
Mean (**A**) dorsal body area (cm^2^ ± SE), (**B**) number of consumed agar blocks during the exposure window of three weeks, and (**C**) dorsal body area (cm^2^ ± SE) during the recovery window of four months of *Rana berlandieri* tadpoles. There were significant differences in the number of consumed agar food blocks across treatments during the recovery period (Rm nested ANOVA: F_3,853_ = 82.45, *p* < 0.0001). There were no significant differences in the number of food blocks consumed between control and Roundup-exposed tadpoles (mean ± SE consumed blocks: control: 2.35 ± 0.17, Roundup: 1.94 ± 0.17, *p* = 0.34), neither between antibiotic- and combination-exposed tadpoles (mean ± SE consumed blocks: antibiotic: 1.10 ± 0.15, combination: 1.36 ± 0.15, *p* = 0.44). However, control tadpoles consumed more food than antibiotic- and combination-exposed tadpoles (*p* < 0.0001, *p* = 0.001, respectively).

**Figure 2 biology-12-01171-f002:**
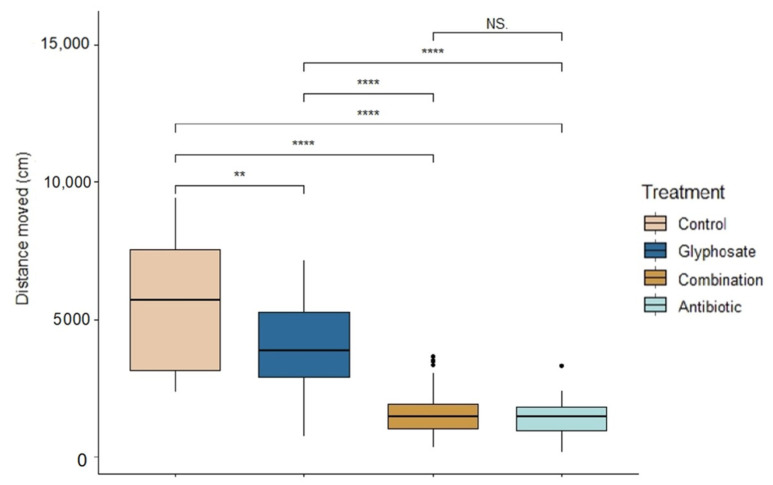
Activity of *Rana berlandieri* tadpoles across four treatments. Box plots indicate median, range, first, and third quartiles. Dots indicate outliers. Asterisks indicate significant differences (Tukey’s HSD comparisons: significance levels ****: 0.001, **: 0.01, NS: not significant).

**Figure 3 biology-12-01171-f003:**
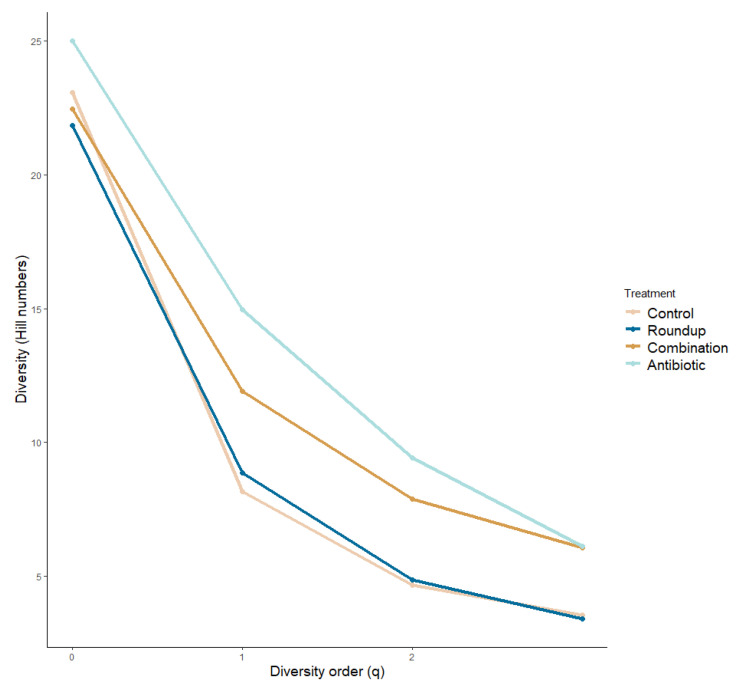
Alpha diversity profile of gut microbiome of *Rana berlandieri* tadpoles exposed to three treatments. There is a greater number of rare species driving the community composition in the antibiotic-exposed tadpoles compared to the rest of the treatments.

**Figure 4 biology-12-01171-f004:**
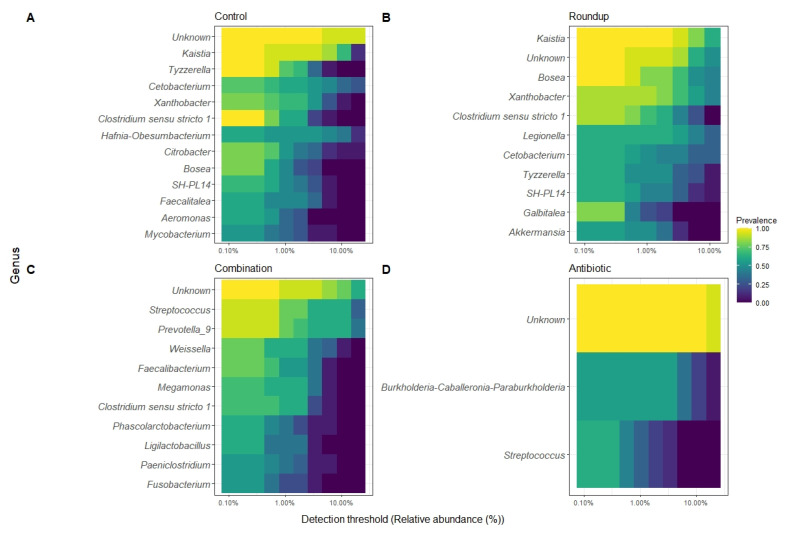
Heatmap of 50% core gut microbiome of (**A**) control-, (**B**) Roundup-, (**C**) combination-, (**D**) antibiotic-exposed *Rana berlandieri* tadpoles. Each row represents a relative abundant ASV in the core microbiome. Cells illustrate the prevalence of each ASV within the core microbiome relative abundance. Cell colors range from blue (low prevalence) to yellow (high prevalence) based on the detection threshold (x axis). We plotted most relative abundant phylum (Figure 5A) and genera (Figure 5B) presented in average across treatments. We identified 18 ASVs that significantly changed in relative abundance across treatments (Appendix A). Proteobacteria and Firmicutes were the most abundant across all treatments. Unknown phyla were found in combination and antibiotic treatments in high relative abundance. For the genera, most noticeable is the high relative abundance of *Legionella* in the Roundup treatment compared to the others (Kruskal-Wallis X^2^_3_ = 29.137, *p* = 2.10 × 10^−6^, Appendix A), as well as the low relative abundance of *Xanthobacter* in combination and antibiotic treatments (Kruskal-Wallis X^2^_3_ = 38.42, *p* = 2.30 × 10^−8^, Appendix A). *Prevotella_9* was found in high abundance only in the combination-exposed tadpoles (Kruskal-Wallis X^2^_3_ = 32.63, *p* = 3.91 × 10^−7^, Appendix A).

**Figure 5 biology-12-01171-f005:**
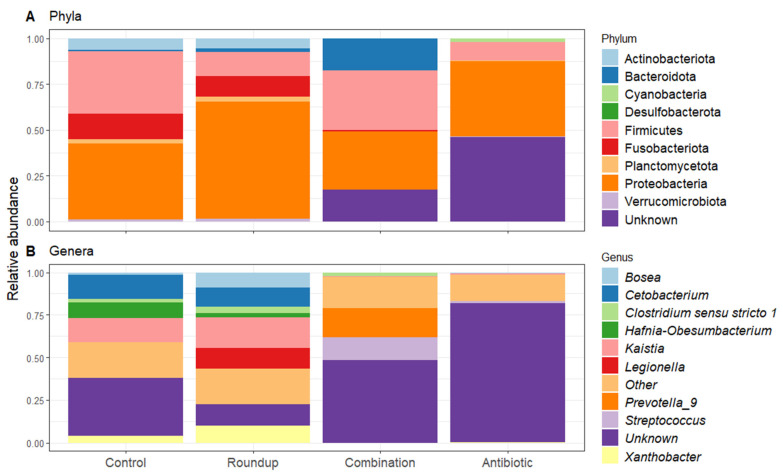
Relative abundance of (**A**) phyla and (**B**) genera found across gut microbiome of *Rana berlandieri* exposed to three treatments compared to control tadpoles. All treatments presented a high relative abundance of Proteobacteria and Firmicutes.

**Table 1 biology-12-01171-t001:** Pairwise comparisons of permutated redundancy analysis (RDA) to analyze the differences in composition and relative abundance of microbiome in the guts of *Rana berlandieri* tadpoles between treatments.

Treatment Comparison	F	*p*	Adjusted R^2^
Control–Roundup	5.725	0.001	0.14
Control–Antibiotic	9.986	0.001	0.23
Control–Combination	12.49	0.001	0.30
Roundup–Antibiotic	4.643	0.001	0.13
Roundup–Combination	7.896	0.001	0.20
Antibiotic–Combination	4.114	0.001	0.10

## Data Availability

Data will be available from the Figshare Digital Repository.

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
