# Peer review of "Exposure to Roundup and Antibiotics Alters Gut Microbial Communities, Growth, and Behavior in Rana berlandieri Tadpoles"

_biology, 2023, doi:10.3390/biology12091171_

Round 1

Reviewer 1 Report (Previous Reviewer 1)

Comments to the Author

I consider that the work entitled “Exposure to Roundup and antibiotics alters gut microbial communities, growth, and behavior in Rana berlandieri tadpoles” is acceptable for publication with minor corrections, which are detailed in the text (pdf file).

-Line 3, in the title and all the text: Please review and update, if necessary, the genus and species throughout the text (https://amphibiansoftheworld.amnh.org/).

-Línea 110-117: Este párrafo debe insertarse en la introducción

-Line 122-123: Line 122-123: Delete "tadpole rearing methodology" and add a full stop.

-Line 149-151: Delete this sentence.

-Linea 346: Standardize citations, according to journal standards, throughout the manuscript.

-Line 368: Delete  "Rana" and add "R."

-Line 412: Delete "chinensis", add "chensinensis"

Author Response

I consider that the work entitled “Exposure to Roundup and antibiotics alters gut microbial communities, growth, and behavior in Rana berlandieri tadpoles” is acceptable for publication with minor corrections, which are detailed in the text (pdf file).

-Line 3, in the title and all the text: Please review and update, if necessary, the genus and species throughout the text (https://amphibiansoftheworld.amnh.org/).

  • We chose not to change the Genus of this species. Despite Frost et al. (2006) recommending the use of the name Lithobatesfor North American Rana. However, we use Rana because  Hillis (2007) (https://doi.org/10.1016/j.ympev.2006.08.001) and Yuan et al. (2016)  returned all Lithobates to Rana, based on the clear monophyly of a larger group that include these genera.

-Línea 110-117: Este párrafo debe insertarse en la introducción

  • Moved to the last paragraph of the discussion

-Line 122-123: Line 122-123: Delete "tadpole rearing methodology" and add a full stop.

  • Deleted

-Line 149-151: Delete this sentence.

  • Deleted

-Linea 346: Standardize citations, according to journal standards, throughout the manuscript.

  • Fixed

-Line 368: Delete  "Rana" and add "R."

  • Fixed

-Line 412: Delete "chinensis", add "chensinensis"

  • Fixed

Reviewer 2 Report (Previous Reviewer 3)

Major inconsistencies found in the earlier version seem to have been properly fixed.

Please check the following:

L114: "Rana" should be in italics

L115: "deeper - tails". Check hyphen usage

L123: full stop missing after "methodology"

L147: " we gently and caught". Check clarity, missing words?

L366: check en dash usage before "86"

Author Response

Major inconsistencies found in the earlier version seem to have been properly fixed.

Please check the following:

L114: "Rana" should be in italics

  • Fixed

L115: "deeper - tails". Check hyphen usage

  • Fixed

L123: full stop missing after "methodology"

  • Fixed

L147: " we gently and caught". Check clarity, missing words?

  • Fixed to “gently caught”

L366: check en dash usage before "86"

  • Fixed

Reviewer 3 Report (Previous Reviewer 4)

Comments about the manuscript:

“Exposure to Roundup and antibiotics alters gut microbial communities, growth, and behavior in Rana berlandieri tadpoles”

The gut microbiome is involved in several physiological aspects ranging from digestion to defense against pathogens. Microbiome knowledge can thus be used to assess the health of the host. Amphibians, fragile species and their microbiome are very sensitive to pollution, in particular by antibiotics. The aim of this work was to research the role of the intestinal microbiome on the physical condition and the phenotype of tadpoles of Rana berlandieri taken as a model. To do this, the authors studied tadpoles exposed to Roundup®, to a cocktail of antibiotics and to a mixture of Roundup and antibiotics, compared to a control group.

This important work focuses on the effects of pollutants on amphibians, fragile species, by considering the effects on the microbiome, which is an approach that is still not widely used and which is of definite interest. This study deserves to be published after some improvements to the manuscript. Here are some remarks.

Page 3, line 128. “We reared the tadpoles until Gosner stage 25”: give the reference (56) here instead of page 4 (don't forget to modify the numbering of the references).

Page 3, line 131. “We exposed R. berlandieri tadpoles to four treatments for three weeks:”: Specify how many tadpoles were used for each treatment or control.

Page 4, lines 165-166. “we used the four tadpoles from each tank that we did not dissect after the exposure period ended.”: a table with the number of tadpoles used at the different stages of the experiment would be useful to clarify the text.

Page 5, Results: from line 228: it would seem useful to me to give a title to the paragraphs devoted to 1) the consumption of agar food blocks and the dorsal zone of the body of tadpoles; 2) tadpole activity; 3) to the intestinal microbiota.

Page 9, line 312. “detection threshold (x axis).We plotted most relative abundant phylum (Figure 5a) and genera”: Isn't there a mix up between figure 4 legende text and main text? Please check and correct if necessary.

References: check references and see if they are presented according to the journal's standards. Look at the website: https://www.mdpi.com/journal/biology/instructions

Author Response

The gut microbiome is involved in several physiological aspects ranging from digestion to defense against pathogens. Microbiome knowledge can thus be used to assess the health of the host. Amphibians, fragile species and their microbiome are very sensitive to pollution, in particular by antibiotics. The aim of this work was to research the role of the intestinal microbiome on the physical condition and the phenotype of tadpoles of Rana berlandieri taken as a model. To do this, the authors studied tadpoles exposed to Roundup®, to a cocktail of antibiotics and to a mixture of Roundup and antibiotics, compared to a control group.

This important work focuses on the effects of pollutants on amphibians, fragile species, by considering the effects on the microbiome, which is an approach that is still not widely used and which is of definite interest. This study deserves to be published after some improvements to the manuscript. Here are some remarks.

Page 3, line 128. “We reared the tadpoles until Gosner stage 25”: give the reference (56) here instead of page 4 (don't forget to modify the numbering of the references).

  • Fixed

Page 3, line 131. “We exposed R. berlandieri tadpoles to four treatments for three weeks:”: Specify how many tadpoles were used for each treatment or control.

  • In the prior paragraph we explained that we had 6 tadpoles per tank. But to clarify, we added n = 48 for each treatment as suggested.

Page 4, lines 165-166. “we used the four tadpoles from each tank that we did not dissect after the exposure period ended.”: a table with the number of tadpoles used at the different stages of the experiment would be useful to clarify the text.

  • In the prior paragraph we clarified that “We collected the guts of two tadpoles per treatment tank (n = 16/treatment)”
  • Here we added that we had n = 32/ treatment.

Page 5, Results: from line 228: it would seem useful to me to give a title to the paragraphs devoted to 1) the consumption of agar food blocks and the dorsal zone of the body of tadpoles; 2) tadpole activity; 3) to the intestinal microbiota.

  • We added 1 Dorsal Body Area and Food Consumption, 3. 2 Behavior, and 3.3 Gut Microbiome

Page 9, line 312. “detection threshold (x axis).We plotted most relative abundant phylum (Figure 5a) and genera”: Isn't there a mix up between figure 4 legende text and main text? Please check and correct if necessary.

  • The legends are correct but there was no space in the pdf making it confusing. Also we added more clarification in the text for figure 4 “We described the detection threshold by the core gut microbiome of tadpoles from each treatment (Figure 4a-d)”

References: check references and see if they are presented according to the journal's standards. Look at the website: https://www.mdpi.com/journal/biology/instructions

  • We have redone the references and the numbering

This manuscript is a resubmission of an earlier submission. The following is a list of the peer review reports and author responses from that submission.

Round 1

Reviewer 1 Report

The authors in this work explored how glyphosate alters gut bacterial community composition of Rana berlandieri, tadpoles. They disrupted the tadpole gut microbiome with an antibiotic cocktail and a combination of glyphosate and antibiotic to see the interaction between the two treatments. They also explored the relationship between the microbial communities and behavior, growth, and survival in Rana berlandieri, tadpoles.

The approaches are scientifically sound and the methods are appropriate for the objectives as stated.

Some minor corrections are detailed in the text (pdf document).

Reviewer 2 Report

Villatoro-Castañeda et al present a study that investigates the influence of the glyphosate and antibiotics on the gut microbial communities from tadpoles of the frog Rana berlandieri. They also evaluated the impact of this herbicide in the behavior, growth, and survival of tadpoles. The authors found that gut microbial communities (overall and core bacterial community) changed across treatment in terms of alpha and beta diversity. The authors conclude that the gut microbial community on tadpoles of R. berlandieri are sensitive to glyphosate and can also impact host behavior and growth.

I do think that the manuscript could use more clarity and falls short in evaluating the effect of the herbicide on the tadpoles themselves. The authors fail demonstrate a clear relationship of the microbial communities disrupted by the herbicide and antibiotics with the tadpole’s growth and behavior. They also fail to recognize/justify why they are investigating the role of antibiotics. Also, some ideas are confusing and difficult to understand what they meant. I provide a list of specific comments that may help address the issues I’ve highlighted, among other comments.

Line 11: Why use the term healthy? Do you have a background of the gut microbiome that makes you affirm you have a healthy microbiome? You should be careful with the use of this term.

Line 18: you are saying that tadpoles are sensitive to environmental pollutants, but you don’t have evidence to say that in a broad sense. You are only evaluating a herbicide.

Line 20: Why conservation physiology of amphibians? I don’t think that is an area of conservation. Maybe remove physiology and just focus in conservation of amphibians.

Line 23: Please remove healthy

Line 24: Should be … by comparing tadpoles of Rana berlandieri

Line 24-2: Please add “with”: “… by comparing Rana berlandieri in a control group with tadpoles exposed to…”

Line 30: Please be more specific of which treatments were different between them.

Line 41: Please add: horizontal/vertical

Line 42: Please remove composition, when you mention diversity you are being explicit of alpha and beta diversity, and beta diversity involves the composition.

Line 43: Please change to … that influence host microbial diversity is not well-known.

Line 44: Remove “Aditionally”

Line 44: Remove baseline and add “natural range of variation”

Line 62: Please add a reference after disturbed.

Line 70-72: This last sentence seems to be out of context. Why talking about amphibian declines now? I don’t think is necessary to mention it, I would just remove the sentence. You are not linking amphibian declines with pesticides in this manuscript. You could mention something about it in the discussion, but again I don’t think it is necessary.

Line 74: You are saying “non-lethal”, do you have any data of the LC50 for your species or other related species? that makes you assure the concentrations that you are using are non-lethal for your study species

Line 76: It is not clear up to this point a justification of why the authors are using antibiotics in this study and especially in combination with herbicides. This would be critical for the study. Also, there is no context of antibiotics in the introduction, this should be addressed.

Line 79: Would be good to add information about the biology and habitat of Rana berlandieri. Why did authors choose this species for their study?

Line 82-89: I suggest authors should move this paragraph above. Probably after line 72. I am also concern about what the herbicide the authors used for this study. It is not very clear if the authors use the active ingredient glyphosate or the product Roundup. If authors used the product Roundup, they should be aware that this product might have other compounds that could potentially be interreacting with the gut microbiome and tadpoles. They should clarify this part, across the manuscript and title since they are saying across the manuscript that they use glyphosate. Be very careful with the terms that you use.

Line 91: Please remove “half of”. This is confusing.

Line 101: Are these tanks made of glass? Pesticides can stick in plastics affecting concentrations in treatments, and potentially disrupting the observed effects.

Line 106: Why are you using those concentration of antibiotics?

Line 113-115: The ideas in this sentence are confusing. Please improve.

Line 116-117: It is not clear if authors took the animals from the container and took a picture to them, or they just put the ruler on the jar. Please be more specific.

Line 120: Please add citation after Image J.

Line 130: I think it would be a good idea to mention that it is long term experiment (say here 4 months recovery).

Line 134: Please change “dead” to “mortality”.

Line 143: Please add a citation to justify why you used a 0.22 micrometers filter paper.

Line 149: Please remove “several”

Line 150: I am concern about the two PCR cycle for only antibiotic and combination treatment samples. Why did you not were constant of this protocol for all treatments? This could potentially bias your results.

Line 153: remove “analysis”

Line 171: Why ANOVA? For overall significance of the mixed model? For multiple comparison tests? Why not performing a poshoc test of the mixed model? This part is not clear.

Line 175: Please indicate where did you cut your forward and reverse reads.

Line 177-179: This sentence is not clear. Two ideas are combined here, please revise.

Line 180: What is a diversity profile? Please clarify.

Line 182: It is no clear which beta diversity metric(s) was used for the analysis. Please indicate.

Line 183: You cannot calculate the core of a treatment, you can calculate the core of the gut microbial communities. Please be careful how you describe your methods.

Line 184-186: The authors need to be more specific on which multivariate analysis they performed to detect an effect of pesticide and antibiotics to microbial community relative abundance (are you talking about beta diversity here when you use the term microbial community relative abundance?)

Line 186-189: Please improve these two sentences. They are hard to understand. Specifically, be more explicit in how you performed the Source Tracker analysis.

Line 221: Please change to … tadpoles in control treatment consumed more food than tadpoles in antibiotic (p<0.0001) and combination treatments (p = 0.001).

Line 224-225: Compare to what are the p-values related?

Line 229: This is boxplot It shows the median, why are you saying “mean activity”?

Line 237: Which beta diversity metrics are you using here when you talk about microbiome composition and relative abundance?

Line 257-258: You should be more specific of the observed patterns, just saying we plotted the most relative is not a result, is more a description/methods. Please describe relevant patterns you see in there. Perphaps is better to move figure 5 to supplementary material and do a differential abundance analysis with the core microbiome data such as DeSeq2, edgeR or ANCOM2.

Line 260-262: Please indicate what core are we looking at 95% 90% or 50%.

Line 271-273: Just saying we plotted… is more methodological. You should describe the patterns you are finding, not just mentioning figure 6.

Line 286: What means significance here?

Line 287: It is not clear why you are saying the role of the microbiome in overall health and fitness of anuran larvae. I think this is appropriate based in your study.

Line 290: How do you know those are pathogenic bacteria. This is a very strong assumption. Also, this is very new in the MS. You didn’t mentioned anything about pathogenic bacteria in the results section. For me is out of context. Please improve in results.

Line 300: “a trend of smaller glyphosate exposed tadpoles”. This sounds weird, perhaps saying smaller tapdoles exposed to glyphosate.

Line 309: I until this point of the MS that I know why the authors use antibiotic treatment in their study. They should improve this topic in the introduction.

Line 310: Again, are you sure that your tadpoles have a healthy microbiome? Be careful with this strong assumption.

Line 312-317: This is a good justification that can be use in the introduction. Also, is this antibiotic present in the wild environment of the frogs? Why using this antibiotic in the study? It doesn’t have biological sense without a proper explanation.

Line 363-366: The authors don’t have a robust analysis to link the growth and behavior with microbiome. This linkage is very vague in the manuscript. The analyses were done separate. I don’t agree with the conclusion suggested by the authors. Would be ideal to perform a pathway analysis to detect potential linkages between microbiome, growth and behavior.

Reviewer 3 Report

In this paper the authors exposed stage-25 tadpoles of Rana berlandieri to 3 different treatments over a 3-week period: (1) glyphosate (Roundup) at a rather low concentration; (2) antibiotic cocktail; (3) a cocktail of glyphosate and antibiotics. As far as I understand the main aim of the study was to investigate if and how glyphosate alters gut microbial communities. The authors also investigated whether glyphosate alters behavior, growth, and survival.

The paper is interesting but contains several inconsistencies. On the first page (in simple summary), the authors state that the use of antibiotics was to “kill the natural microbiome of the tadpoles”. I don’t really understand how the gut microbiome composition could be altered if it has been previously killed. This should be clarified. It seems to me that the aims of the study changed over time, maybe because of the results (?)

What is the rationale for the Roundup concentration? This concentration seems rather low (and likely decreased until day 10 when the water was changed). What’s the rate of Roundup breakdown? In comparison the antibiotic cocktail seems rather heavy, and the concentrations seem unbalanced between your experiments (high doses of antibiotic vs low doses of Roundup). Please clarify.

I am a bit surprised that no reference is made to the work of Rick Relyea, who tested Roundup on tadpoles (including Rana) and investigated its effects on growth and behavior. There is some redundancy with some of his work.

L14: “to kill the natural”, not “to kill of the natural”

L85-86: growth and food consumption are repeated twice

L103-109: very confusing paragraph. Your first say “we exposed tadpoles to xxx”, then “to do this we exposed tadpoles to 3 treatments”. Also, a parenthesis is missing L104

L111-112: 10h of light, 12h of darkness. Given that there are 24h in a day, what happened during the 2 missing hours?

L125: not clear when you did this, was it after 3 weeks?

L1311: water mentioned twice

Reviewer 4 Report

Comments on the manuscript:

“Exposure to glyphosate and antibiotics alters gut microbial communities, growth, and behavior in Rana berlandieri tadpoles”

The gut microbiome acts on the physiology of the body. It is possible that the presence of pollutants changes this gut microbial activity. The present work concerns the study of the modification of the intestinal microbiome in the anuran amphibian Rana berlandieri. For this, tadpoles were put in contact with glyphosate and a mix of antibiotics to examine the effects of these molecules on the intestinal microbiome by studying various parameters (growth, diet, etc.) and examining their effects on the intestinal microbiome. The bacterial composition of the external environment and the intestine have been studied.

This work is very interesting and brings new data in a field which deserves to be explored more and more. The manuscript could be accepted for publication after some improvements. Here are a few remarks.

Page 3 and following, Materials and methods: the protocol is quite difficult to follow and a table or figure summarizing the experiments would be useful for the reader.

Page 4, lone 111: “light cycle (10 hours light/12 hours darkness)”: the addition is 22 hours, not 24: is it normal? Explain.

Page 6, line 143: “Petri” (with a capital first letter), not “petri” (Petri is the name of a microbiologist).Page 6, line 153: write “Falcon” (with a capital first letter) instead of “falcon”, (it is the name of a supplier).